# Keeping a Secret Requires a Good Memory:
# Unconditional Streaming Lower-Bounds for Differentially Private Algorithms

**Alessandro Epasto** [1]   **Xin Lyu** [2]   **Pasin Manurangsi** [3]

## Abstract

We study the computational cost of differential privacy in terms of memory efficiency. Specifically, we establish for the first time an unconditional space lower bound for user-level differential privacy by introducing a novel proof technique based on a multi-player communication game. We apply our framework, as an example, to the fundamental problem of estimating the number of distinct elements in a stream: we prove that any private algorithm requires almost $\widetilde{\Omega}(T^{1/3})$ space (where $T$ denotes the length of the stream) to achieve certain error rates in a promise variant of the problem, resolving an open problem in the literature (by Jain et al. (Jain et al., 2023a) and Cummings et al. (Cummings et al., 2025)) and establishes the first exponential separation between the space complexity of private algorithms and their non-private $\widetilde{O}(1)$ counterparts for a natural statistical estimation task. Furthermore, we show that this communication-theoretic technique generalizes to broad classes of problems, yielding lower bounds for private medians, quantiles, and max-select.

## 1. Introduction

As artificial intelligence becomes central to daily life, technology companies must process ever-larger amounts of *private* data. To ensure this data is safeguarded, a major effort in machine learning has focused on designing privacy-preserving algorithms that are both *accurate enough* to be useful and *efficient enough* to be deployed in real-world systems. In this context, the gold standard is to design algorithms with strong theoretical guarantees in terms of *user-level* differential privacy (henceforth DP). Celebrated results in this area include efficient and practical algorithms for a multitude of fundamental machine learning and data

analysis problems, including training deep learning models privately (Abadi et al., 2016), private synthetic data generation (Jordon et al., 2018), private statistical estimation (Liu et al., 2022), private combinatorial optimization (Gupta et al., 2010), and many others.

Despite these positive results, researchers frequently encounter problems that elude a private solution with the desired efficiency. For instance, consider the fundamental problem of privately estimating the number of distinct elements in a stream, which serves as a running example in this paper. Prior work established private algorithms using polynomial space in the input stream that are provably nearly-optimal in error (Cummings et al., 2025; Jain et al., 2023a), but left as an open problem whether private algorithms with poly-logarithmic space (and equal accuracy) are possible, akin to the non-private setting.

**Lower bounds on private algorithms.** In such cases, it is desirable to determine, by establishing a provable impossibility result, whether the elusive efficiency goal is actually mathematically incompatible with the privacy requirement. Fortunately, regarding *accuracy*, multiple results have elucidated the inherent tradeoff with privacy, providing general lower bound techniques (e.g., one-way marginal queries Bun et al. (2018); Dwork et al. (2015)) that have been used to bound the accuracy of private algorithms for many problems. As for the *efficiency*, the two most common measures in algorithmic design are: *running time* and *memory*. While there have been a line of work demonstrating that some tasks require more running time with privacy than without (Dwork et al., 2009; Ullman & Vadhan, 2020; Blocki et al., 2025), much less is understood regarding the privacy versus memory tradeoffs.

Specifically regarding memory tradeoffs with privacy—the focus of this work—to the best of our knowledge, the only existing result is the seminal work of Dinur et al. (2023), which provides the first space bound for a private algorithm. This work shows that, under certain cryptographic assumptions, a specifically crafted streaming problem involving encryption requires large space if solved privately (whereas it can be solved non-privately with exponentially less space). While this proves that privacy and low space can be incompatible, the result has a few limitations that our

[1] Google Research, USA [2] UC Berkeley, USA [3] Google Research, Thailand. Correspondence to: Xin Lyu <xinlyu@berkeley.edu>.

*Proceedings of the 43rd International Conference on Machine Learning*, Seoul, South Korea. PMLR 306, 2026. Copyright 2026 by the author(s).

work addresses. First, this bound relies on cryptographic assumptions, whereas it would be ideal to prove unconditional information-theoretic bounds that are inherent to the problem itself. Second, and more importantly, the proof technique relies on artificially constructing a hard problem and does not generalize to standard problems encountered in the real world (e.g., standard statistical estimation problems). Finally, it would be ideal for the bound to convey an intuitive explanation for *why* certain natural private algorithmic problems require large space.

**Our contribution.** Our work addresses these limitations and resolves the open problem posed by prior work (Jain et al., 2023a; Cummings et al., 2025). We introduce a novel proof technique for establishing unconditional space bounds for private algorithms based on designing a novel communication game. We demonstrate that this technique applies to a variety of widely studied natural statistical estimation problems, including privately counting the number of distinct elements in a stream, as well as computing medians, quantiles, and solving the max-select problem (see Section 2 for formal definitions).

Specifically, for the distinct elements problem in the turnstile model[1], we show that the prior work of Cummings et al. (2025) is essentially space-optimal in certain regimes. In a promise version of the streaming distinct count problem, where each user's contribution to the stream is bounded (see Section 2.1 for details), Cummings et al. (2025) show how to design a private algorithm with error $o(T^{1/3})$ with space complexity $\widetilde{O}(T^{1/3})$ [2]. We show that, in order to take advantage of the promise, the space usage of (Cummings et al., 2025) is nearly optimal: any algorithm with comparable accuracy must use a space of $T^{1/3-o(1)}$ bits. This establishes, for the first time, an exponential separation in memory requirements due to privacy for a natural computational problem. We believe our communication game technique could be used to establish bounds for a broader range of problems.

**Intuition: "capping" contributions requires space.** From a practical perspective, our proof technique links the hardness of low-memory private algorithms to a widely observed phenomenon: providing user-level privacy often requires capping the contributions of users to the algorithms. A well-known technique in the design of many privacy-preserving algorithms is, in fact, limiting the number of updates any user can provide to a dataset (this is often referred to as contribution bounding or capping, e.g., Amin et al. (2019)). Doing so is often *sufficient* to achieve strong privacy protections (after adding appropriately scaled noise), as restricting

outlier users from contributing disproportionately to the solution protects their identities from being compromised. Intuitively, keeping track of $k$ "over-active" users that have exhausted their contribution budget requires at least $\tilde{O}(k)$ bits of memory. The crux of our work is showing that, for certain problems, this widely-used (but memory intensive) strategy is not just useful, but essentially necessary. This result is, perhaps, counter-intuitive. Indeed, one could imagine many low-space alternative approaches, for instance based on aggressive down-sampling, where user contributions are effectively limited while using much less space.

To achieve this lower bound, we design a multi-player communication game where each player possesses contributions from a subset of (potentially repeated) users. Each player needs to solve the problem for their subset of data (privately) and then can pass useful (non-private) information to the next player regarding which users (at most $k$ out of $n$) have exceeded their contribution limits. We show that to win such a game, the players need to communicate $\tilde{O}(k)$ bits of information, and we relate this requirement to impossibility results for solving the problem privately.

**Organization.** Before proving our main results (Theorem 2.1), we first briefly review the relevant work. Next, in Section 2, we formally define user-level DP in the streaming setting and present our hardness construction. Finally, in Section 3, we formally prove the main result.

## 1.1. Related Work

Our work spans several areas, including lower bound techniques in DP and data streaming with privacy.

**Lower bounds for private algorithms.** Significant work has been dedicated to understanding the accuracy-privacy tradeoffs in DP. This includes the reconstruction attack (Dinur & Nissim, 2003) and subsequent discrepancy-based lower bounds (Muthukrishnan & Nikolov, 2012; Nikolov et al., 2013); the packing argument (Hardt & Talwar, 2010); the fingerprinting method (Bun et al., 2018; Dwork et al., 2015; Steinke & Ullman, 2017; Lyu & Talwar, 2025); and others (Bun et al., 2015; Acharya et al., 2021; Cohen et al., 2024). The reader is also referred to the comprehensive textbook (Vadhan, 2017) for further discussion on this topic.

To the best of our knowledge, the only known space lower bound separating DP versus non-DP streaming algorithms is that of Dinur et al. (Dinur et al., 2023). In that work, the authors design a novel cryptography-related streaming problem (where encryption keys are received in a certain order), which is then used to prove (under cryptographic assumptions) that a space-bounded algorithm cannot solve the problem privately. In contrast, the goal of our paper is to design lower bounds that are *information-theoretic* (i.e., do not rely on any assumption apart from accuracy)

---

[1] In the turnstile model (Muthukrishnan, 2005), a user can arrive or leave the set at each time step. See Section 2 for a formalization.

[2] Throughout the paper we use the $\tilde{O}$ notation to neglect poly-logarithmic factors in $T$.

and that can be applied to natural problems that have been well-studied in the literature.

**Data streaming algorithms.** The literature on non-private streaming algorithms is vast and we refer interested readers to surveys on the topic, e.g., Muthukrishnan (2005), for more detail. Nevertheless, we point out that the problems we study–counting distinct elements and quantiles–are among the most well studied problems in the area. (See e.g., Kane et al. (2010); Cormode et al. (2023) and references therein.)

Particularly relevant to our work is the area of DP applied to streaming algorithms. Data streaming applications in which one needs to track statistics over a large number of potentially private events occur in many real-world scenarios, including digital advertising, network monitoring, and database systems. This has motivated the growing area of continual release DP algorithms in streaming (Dwork et al., 2010). Beyond the already cited work of Jain et al. (2023a); Cummings et al. (2025) on count distinct elements in the turnstile model, (Bolot et al., 2013; Ghazi et al., 2023) has studied the problem (and its variants) on the more stringent *cash register* model (where users can only join but not leave the set). Recent works Andersson et al. (2026); Aamand et al. (2026) have shown improved the error rate for private count distinct elements but did not focus on improving the space requirements of Cummings et al. (2025). There has also been significant work on related problems such as sum and histogram estimation (Henzinger et al., 2024; 2023), moment estimation (Epasto et al., 2023), and graph statistics (Jain et al., 2024).

## 2. The Problem Setting

Before presenting our main results, we introduce the concept of user-level differential privacy in the context of continual release streams, as well as the problems for which we provide space lower bounds in our work.

**General setting.** All problems we study share the following continual release setting. Consider a set of $N$ users, denoted by $u_1, \ldots, u_N$. The users contribute to a stream $x$ of $T$ steps $x_1, \ldots, x_T$. Each step $t$ of the stream, there is either an empty update ($x_t = \bot$) or the step consists of one user $u_{i(t)}$ submitting a problem specific update $j(t)$ (denoted as $x_t = (u_{i(t)}, j(t))$). After receiving the update $x_t$ the algorithm responds with an output $a_t$ which seeks to approximate the true answer $a_t^*$ for step $t$. The problems (defined below) differ on the specific input and true answer.

**Accuracy requirement.** In all problems we seek the answers of the problem $a_t$ to be a $(\tau, \eta)$-approximation to the true answer $a_t^*$, for all steps $t$, $a_t \in (1 \pm \tau)a_t^* \pm \eta$. To simplify the notation we let $\tau = .1$ and focus only on the additive error $\eta$.

**Privacy constraint.** For all problems we seek to design user-level differentially private algorithms. In user-level DP, a pair of streams $x, x'$ is considered neighbors if one can be derived from the other by replacing *all* updates of a single user $u$ with $\bot$; we then enforce that the output distributions of the algorithm across all $T$ steps on $x$ and $x'$ are $(\epsilon, \delta)$-indistinguishable. Formally, denote by $A(x)$ (resp. $A(x')$) the length-$T$ vector of the responses from the algorithm on input stream $x$ (reps. $x'$). For any event $S$ on the outcome vector, $(\epsilon, \delta)$-user-level DP requires that:

$$\Pr[A(x) \in S] \le \Pr[A(x') \in S] \cdot e^\epsilon + \delta.$$

**The CountDistinct problem.** We can now define the first problem formally. For the CountDistinct problem the updates of the stream $x_t$ are either empty ($x_t = \bot$) or consists of one user $u_{i(t)}$ submitting either a "+1" or a "−1" (denoted as $x_t = (u_{i(t)}, \pm 1)$). At any time step $t \in [T]$, we can define the frequency vector $f^t \in \mathbb{R}^N$, where $f_i^t$ denotes the total sum of the updates from user $u_i$ up to time $t$ (users with no updates are assumed to have a 0 sum). Our lower bound holds for an even more restrictive version of the problem where the frequency $f_i^t \in \{0, 1\}$ for all users $u_i$ and all times $t$, making the lower bound stronger.

The CountDistinct problem requires the algorithm to output, after each step $t$, a value $a_t$ that approximates $a_t^* = \|f^t\|_0$ (i.e., the number of users with a non-zero sum at time $t$).

Notice that like in prior work (Cummings et al., 2025), this formulation allows both additive and multiplicative errors in the estimate. The additive error is necessary given the strong information-theoretic lower bounds against differentially private algorithms (see, e.g., (Jain et al., 2023b;a)). The multiplicative error is known to be required to design space-efficient streaming algorithms for this problem (such as HyperLogLog, MinHash, etc.) (Alon et al., 1996; Kane et al., 2010), even in the non-private setting. Observe how the formulation naturally models, for instance, counting how many users are currently active at any time (where a +1 update indicates the user activates, and a −1 update indicates the user deactivates, at a certain time).

**The MaxSelect problem.** In the MaxSelect problem (Jain et al., 2023b), there is a publicly known set of $d$ features. The input of every user is a binary vector $v_u \in \{0, 1\}^d$, which is initialized to the all-zero vector and users are allowed to update their input by flipping one bit at a time. More formally at step $t$, the algorithm receives an update that is either empty ($x_t = \bot$) or consists of one user $u_{i(t)}$ submitting the flipping (from 0 to 1 or vice versa) of a position $j(t)$ of the their vector (denoted as $x_t = (u_{i(t)}, j(t))$).

At every time step $t$, the streaming algorithm needs to track approximately the most "popular" feature among the $d$ choices. More precisely, let $\sum_u v_{u,j}^t$ be the value of item

$j$ at time $t$. Denote by $j^*(t)$ the feature with true maximum value at time $t$. With slight abuse of notation we let $a_t^* = \sum_u v_{u,j^*(t)}^t$ be the true answer. The algorithm seeks to output an item $a_t$ such that $\sum_u v_{u,a_t}^t$ is a $(\tau, \eta)$-approximation to $a_t^*$.

An example instantiation of MaxSelect may be the following: a book lovers platform hosts $d$ books, every time a user starts (or completes) reading a book corresponds to a flip for that user and book. The platform wants to track the "most popular" book being read at the time.

**The Quantile problem.** There is a publicly known and ordered universe $[U] := \{1, \ldots, U\}$. The input of every user is an item $v_u \in [U]$ and the inputs may change over the stream. Specifically, at time step $t$, the algorithm receives an update that is either empty ($x_t = \perp$) or consists of one user $u_{i(t)}$ changing their item to $v(t)$ (i.e. $x_t = (u_{i(t)}, v(t))$).

Furthermore, at every time step, the streaming algorithm needs to answer a quantile query: that is, a rank parameter $0 \le k \le N$ is specified and the algorithm needs to respond with an item $a_t \in U$ such that the rank of $a_t$ relative to the collection of users' items is approximately $k$. Namely, denote $\hat{k} = |\{i \in [N] : v_{u_i} \le a_t\}|$ as the rank[3] of $a_t$ relative to $\{v_{u_1}, \ldots, v_{u_N}\}$. We require that $\hat{k}$ to be a $(\tau, \eta)$-approximation to $k$. We also note that in this problem formulation, the query parameter $k$ is considered a piece of public information, and the algorithm does not need to protect its privacy (as otherwise the problem is impossible to solve).

We note that both problems admit differentially-private algorithms with $\widetilde{O}(T^{1/3})$ additive error (albeit with large, polynomial space), as well as a $\text{poly}(\log(n), d, \log(U))$ space non-private streaming algorithms with small constant multiplicative error (see for example (Cormode et al., 2023; Jain et al., 2023a; Cummings et al., 2025)).

## 2.1. CountDistinct: the occurrence bounding technique

We henceforth focus only on the CountDistinct problem which serves as the main example for our lower bound technique. To gain insights on the hardness of the CountDistinct problem we will first review the approaches used in prior work to achieve polynomial space algorithms for this problem. The intuition gained from this problem will serve to understand the lower bound in all other cases.

For this problem without any restriction on user contributions to the stream, with user-level differential privacy, it is known (Jain et al., 2023a) that the minimax (additive)

---

[3]If there is one or more $v_{u_i}$'s that are of the same as $a_t$, the rank of $a_t$ among those $v_{u_i}$'s can be *arbitrarily* chosen for the purpose of evaluating the correctness.

error rate of the problem scales as $\widetilde{\Theta}(T^{1/3})$ (for an arbitrary small constant multiplicative approximation). The prior work of (Cummings et al., 2025) achieves this error rate, with high probability, on arbitrary streams, with $\widetilde{\Theta}(T^{1/3})$ space. Furthermore, (Cummings et al., 2025) devises an improved algorithm for a promise variant of the problem that imposes a cap on the contributions of the users to the stream. This is formalized through the *occurrency* concept, which we will discuss next. (See also Jain et al. (2023a) for the related concept of flippancy.)

**Occurrency of the stream.** The *occurrency* of a user $u_i$ in a stream is defined as the number of updates involving that user. The stream is $(w, k)$-*occurrency bounded* if the stream contains at most $k$ users with strictly more than $w$ updates. This definition is inspired by the well-known long-tailed user distribution observed in practice, where it is known that a small number of users are significantly more active than the vast majority (Mahanti et al., 2013; Clauset et al., 2009).

Prior work (Jain et al., 2023a; Cummings et al., 2025) demonstrated that if the stream is guaranteed to be $(w, 0)$-occurrency bounded (i.e., no user has more than $w$ updates), an additive error of $O(w^{1/2} \cdot \text{polylog}(T))$ is achievable (with high probability).

**Capping the occurrency of a stream.** Real-world datasets, however, are rarely strictly $(w, 0)$-bounded; it is common for a few users (say $k$) to have an occurrency higher than $w$. A natural approach to solving these instances, used in (Cummings et al., 2025), is the following: first identify "heavy" users with more than $w$ updates, then ignore all future updates from such users. By effectively capping all users at $w$ updates, we obtain a $(w, 0)$-bounded stream. Consequently, in a $(w, k)$-occurrency bounded stream, assuming a capping at $w$ occurrency that removes the $k$ heavy users, we can obtain an additive error of $O(w^{1/2} \cdot \text{polylog}(T)) + k$.

Note that any arbitrary stream is $(T^{2/3}, T^{1/3})$-occurrency bounded, as at most $k = T^{1/3}$ users can have $w = T^{2/3}$ updates (given that the total number of updates is $T$). Based on this, Cummings et al. (2025) use (an approximate version of) the aforementioned capping technique to obtain an algorithm with an additive error of $\tilde{O}(w^{1/2} + k) = \tilde{O}(T^{1/3})$ and also $\tilde{O}(T^{1/3})$ space for arbitrary input streams.

**Space complexity of occurrency bounding.** The challenge with this approach is that the set of heavy users is not known to the algorithm a priori, and their IDs need to be identified on the fly. Intuitively, this requires a large amount of memory. An obvious solution is to maintain a counter for each user to track their number of updates. Whenever a user's counter crosses a predetermined threshold, the algorithm marks that user as heavy and ignores them in subsequent analysis. More sophisticated algorithms can use sampling or *sketch*-based solutions to estimate these counts and iden-

tify "heavy hitters" (e.g., the famous CountSketch (Charikar et al., 2002)). Nonetheless, all known approaches seem to require $\Omega(k)$ bits of working memory—after all, it takes at least $\Omega(k)$ bits simply to remember the set of identified heavy users. In fact, as shown in Cummings et al. (2025), identifying all users with more than $w$ updates requires at least $O(T/w)$ space even allowing some errors.

## 2.2. Our main result

While prior work already showed that occurrency bounding is space-inefficient, this is not sufficient to establish an unconditional space lower bound for our problem (and others). Indeed, a better solution to CountDistinct may not involve at all identifying heavy users.

Our main results show that the aforementioned overhead in memory is actually *necessary*, regardless of whether one wants to explicitly employ the occurrency bounding technique or not. More formally, we will show that any user-level DP algorithm with additive error smaller than $(kw)^{0.5-\Omega(1)}$ must use $(kw)^{\Omega(1)}$ bits of memory on $(w,k)$-occurrency bounded CountDistinct streams. This should be contrasted with the information-theoretical upper bound (Cummings et al., 2025), where we know $\widetilde{O}(\sqrt{w}+k)$ additive error is achievable using a polynomial-space algorithm. Namely, we exhibited a notable information-computation gap on the space complexity of the CountDistinct problem.

We can now state our main theorem[4].

**Theorem 2.1.** *Let $\gamma_w, \gamma_k, \gamma_h \in (0,1)$ be such that $\frac{1}{2}\gamma_w \leq \gamma_k \leq \gamma_h < \gamma_w$ and $3\gamma_h < 1$, and $T \geq 1$ be sufficiently large. Let $w = T^{\gamma_w}, k = T^{\gamma_k}, h = T^{\gamma_h}$. Suppose $A$ is an $(\varepsilon = 0.5, \delta = \frac{1}{T^2})$-DP algorithm for the streaming **Count-Distinct** problem on $(w,k)$-occurrency bounded streams with the following accuracy guarantee: for any fixed stream $x$, with probability $1-\frac{1}{T^2}$ it holds that the responses of $A(x)$ are $(\tau = \frac{1}{10}, \eta = h)$-approximations to the true answers across the stream. Then, $A$ must use a space of at least $M \geq \widetilde{\Omega}\left(\frac{kw}{h^2}\right) = \widetilde{\Omega}\left(T^{\gamma_w+\gamma_k-2\gamma_h}\right)$ bits.*

We interpret the result as the following lower bound: with $k^{o(1)}$ bits of memory, the best achievable error must be off from the best upper bound by a polynomial factor. For example, consider the regime that $\sqrt{w} \approx k$. The known private algorithm Cummings et al. (2025) solves the problem in this regime with an additive error of $O((\sqrt{w}+k) \cdot \text{polylog}(T)) = \widetilde{O}(k)$ with $\widetilde{O}(k)$ space. By Theorem 2.1 we know that any algorithm obtaining an error $h \leq \sqrt{kw} \approx k^{3/2}$ must use a space lower bound of $k^{\Omega(1)}$.

One notable corollary of Theorem 2.1 is the following: let

---

[4]For the ease of exposition, we have fixed the privacy parameters to a common and standard choice of $(\varepsilon = 0.5, \delta = \frac{1}{T^2})$ and we do not focus on how our results scale with $\varepsilon, \delta$.

$\alpha > 0$ be sufficiently small. By setting $\gamma_w = \frac{2}{3}-4\alpha, \gamma_k = \frac{1}{3}-2\alpha$ and $\gamma_h = \frac{1}{3}-\alpha$, we obtain a lower bound that is of order $\widetilde{\Omega}(T^{1/3-4\alpha})$. Formally:

**Corollary 2.2.** *Let $\alpha \in \left(0, \frac{1}{9}\right)$. Any $(\varepsilon = 0.5, \delta = \frac{1}{T^2})$-DP algorithm for **CountDistinct** on $(w = T^{2/3-4\alpha}, k = T^{1/3-2\alpha})$-occurrency bounded stream that provides $(\tau = \frac{1}{10}, \eta = T^{1/3-\alpha})$-approximation guarantee with probability $1-\frac{1}{T^2}$ must use a space of $\widetilde{\Omega}(T^{1/3-4\alpha})$ bits.*

One can use the algorithm from (Cummings et al., 2025) to solve CountDistinct on $(w = T^{2/3-4\alpha}, k = T^{1/3-2\alpha})$-occurrency bounded stream with error $\widetilde{O}(\sqrt{w}+k) \approx T^{1/3-2\alpha}$. The algorithm from (Cummings et al., 2025) uses $T^{1/3+O(\alpha)}$ bits of space, while Corollary 2.2 says that any algorithm with accuracy better than $T^{1/3-\alpha}$ must use $T^{1/3-4\alpha}$ bits of space, demonstrating the algorithm from (Cummings et al., 2025) is nearly tight in this regime.

Other than this choice of $h, k, w$, the trade-off we obtained in Theorem 2.1 is not necessarily tight. Finding a tight trade-off between memory and accuracy on any bounded-occurrency stream remains an interesting open problem.

**Extension to more streaming problems.** We present similar results for the MaxSelect and Quantile problems.

**Theorem 2.3.** *Let $T \geq 1$, $w = T^{\gamma_w}, k = T^{\gamma_k}, h = T^{\gamma_h}$ and $\varepsilon = 0.5, \delta = \frac{1}{T^2}$ be set up the same way as in Theorem 2.1. Assume $A$ is an $(\varepsilon, \delta)$-DP algorithm that for $(w,k)$-occurrency bounded streams that achieves $(\tau = \frac{1}{10}, \eta = h)$-approximation guarantee with probability $1-\frac{1}{T^2}$ for **MaxSelect** (resp., **Quantile**). If $d \geq 2$ for **MaxSelect** (resp., if $U \geq 2$ for **Quantile**), then the algorithm $A$ must use a space of at least $M \geq \widetilde{\Omega}(T^{\gamma_w+\gamma_k-2\gamma_h})$ bits.*

We note that in Theorem 2.3, we only need the problems to be just slightly non-trivial (i.e., $d \geq 2$ for MaxSelect and $U \geq 2$ for Quantile) to design our lower bounds.

The rest of the section is focused on introducing the hard instance used in the proof of the theorem.

## 2.3. Construction of Hard Instances

Here we design our hard instances for the CountDistinct problem, noting that similar ideas apply to two other example problems with minimum modification. Let $w, k, h$ be such that $\sqrt{w} \leq k \leq h \leq T^{1/3}$. Our goal is to construct a collection of $(w,k)$-occurrency bounded input streams, such that any low-space DP algorithm solving the Count-Distinct estimation problem on those streams must incur a large error.

We will consider the following construction of hard instances. Let $[N]$ be the universe of users and select a subset $C \subseteq [N]$ of size $k$ uniformly at random. Elements of $C$ will be understood as the set of heavy users. Let $P = \frac{T}{2w(3h+k)}$.

From our choice of $w, k$ and $h$, we obtain that $P \gg \frac{h^2}{w}$, a crucial condition that we need in the analysis.

Let $S_1, \ldots, S_P$ be a collection of $P$ *disjoint* subsets of $[N] \setminus C$, each of size $3h$, all randomly drawn.

**The stream.** We will construct a stream of length $T = 2P \cdot w(3h + k)$. Specifically, for every *phase* $i = 1, \ldots, P$ and every *repetition* $j = 1, \ldots, w$, we do the following:

1. Nature samples $p(i, j) \sim [0, 1]$. Then, for every $u \in S_i \cup C$, Nature samples a bit $b(u, i, j) \sim \mathrm{Ber}(p(i, j))$ independently (i.e., $b(u, i, j) = 1$ with probability $p(i, j)$ and 0 otherwise).

2. Consider all users from $S_i \cup C$. Process them in the increasing order of IDs: for each user $u$ such that $b(u, i, j) = 1$, insert an update $(u, +1)$. For every $u$ such that $b(u, i, j) = 0$, insert an update $\perp$.

3. After all the insertions are completed, query the algorithm $A$ for the estimate of the current distinct count, let $r(i, j)$ be the output of the algorithm, which we assume has been truncated to the range $[0, 3h + k]$. We also denote $a(i, j) = \frac{r(i,j)}{3h+k} \in [0, 1]$ as the normalized output count.

4. Revoke (i.e., remove) all insertions incurred in Step (2) in the *same order* in which they were inserted.

**Intuition.** Here we explain why the construction is hard against a low-space private algorithm. Essentially, we think of the construction as consisting of $P$ stages, where in each stage the stream asks $w$ random *one-way marginal* queries over users from $C \cup S_i$: namely, for $i \in [P]$ and $j \in [w]$, imagine we create an attribute for every user $u \in C \cup S_i$. Step (1) of the construction means that we set the attribute to "one" with probability $p(i, j)$, independently for every $u \in C \cup S_i$. Step (2) inserts all users that have this attribute as "one". Step (3) then queries for the current distinct count, which exactly corresponds to the number of users with a "one" attribute. Lastly, Step (4) revokes the updates we did in Step (2) so that the algorithm can prepare for the next one-way marginal query.

One can quickly verify that our construction indeed produces a $(2w, k)$-occurrency bounded stream: except for the users from $C$ (there are $k$ of them), every other user $u \in S_i$ participates in at most $w$ one-way marginal queries, where each query needs them to send at most two updates (one insertion and one deletion).

Now, in order to answer the queries up to additive error $h$, one can, for example, simply use the Laplace noise addition mechanism. Here, only counting users from the group $S_i$ yields almost equally good estimates: the additional error incurred is at most $|C| \leq k$. In order to preserve privacy,

this is the preferred option: if the algorithm were to take the users from $C$ into account in every stage, the privacy of $C$ will be quickly compromised after a few stages. However, in order for the algorithm to *not* count elements of $C$, it seems necessary for the algorithm to be *aware* of the identities of $C$. Since we have been careful in implementing Step (2) to hide $C$ among $C \cup S_i$, it seems the only viable option for the algorithm is to first *learn* the set $C$ through the first few stages, and store it in the memory so that the algorithm can intentionally ignore their contribution in subsequent analyses. Notice that the storage of $C$ requires a large memory as it is a random large subset of $[N]$. Our goal is to show that the algorithm has no other low space alternative.

## 3. Proof of Main Results

In this section, we prove our main results by analyzing the construction from Section 2.3.

Our proof strategy is as follows. In Section 3.1, we introduce a communication game called AvoidHeavyHitters and proves its communication complexity lower bound. Then, in Section 3.2, we show that any accurate low-space DP algorithm yields low-communication protocol for the game. Finally, in Section 3.3, we put these together to derive Theorem 2.1. (All missing proofs are deferred to the appendix.)

### 3.1. Multi-Player Communication Game of Avoiding Heavy Hitters

The following abstract communication game will play a crucial role in our analysis.

**The AvoidHeavyHitters game.** Let $w, k, h, P, N$ be as above, and $M, \epsilon_1, \epsilon_2$ be additional parameters. Consider the following communication game with $P$ players: we generate subsets $C, S_1, \ldots, S_P$ as described in Section 2.3. Then, the $i$-th player gets the unordered set $C \cup S_i$ as their input. The $P$ players jointly play a game with a referee: starting from $i = 1$ in order, the $i$-th player receives a message from player $(i - 1)$ (if $i > 1$) and then needs to send a set $Y_i \subseteq S_i \cup C$ of size at least $(\frac{1}{2} + \epsilon_1) \cdot |S_i \cup C|$ to the referee. After that, the player passes a message of length $M$ bits to the next player $(i + 1)$, if $i < P$. The game concludes after the $P$-th player has committed their subset to the referee. The players jointly win the game if there is no $u \in C$ that appears in more than $(\frac{1}{2} + \epsilon_2)P$ submitted subsets.

Observe how the game is quite abstract and does not (explicitly) refer to the CountDistinct problem. For this reason believe this technique can adapt multiple problems beyond the ones studied in the paper. Also notice how we do not impose any restrictions on the protocol in terms of resources (memory, time) except for the size of the messages ($M$).

We note that, if $\epsilon_1 = \epsilon_2$, then a simple strategy here is, for each player $i$, to simply pick a random subset $Y_i$ from $S_i \cup C$ of size $(\frac{1}{2} + \epsilon_1) \cdot |S_i \cup C|$. This baseline protocol requires no communication at all. Intuitively, for larger $\epsilon_1 > \epsilon_2$, the game becomes more challenging as this strategy fails and each player must pass on some information to help the next player (partially) identify $C$. Through a careful information theoretic arguments, we confirm this intuition and prove that the communication has to grow as $\Omega\left(k(\epsilon_1 - \epsilon_2)^2\right)$:

**Lemma 3.1.** *Consider the AvoidHeavyHitters game with $P$ players, and parameters $h, k \geq 1$ and $\epsilon_1, \epsilon_2 \in (0, 1)$ such that $|S_i| = 3h$, $|C| = k$. Then, in order for the players to win the game with probability $1 - \frac{1}{10P}$, the communication complexity must satisfy*

$$M \geq \log \binom{3h+k}{k} - \log \binom{(3h+k)(\frac{1}{2}+\epsilon_1)}{(\frac{1}{2}+\epsilon_2)k}$$
$$- \log \binom{(3h+k)(\frac{1}{2}-\epsilon_1)}{(\frac{1}{2}-\epsilon_2)k} - O\left(\log(h+k)\right).$$

*When $h \geq k$ and $\epsilon_1, \epsilon_2 \in (\frac{1}{k}, \frac{1}{10})$, we may use Stirling's approximation to arrive at*

$$M \geq \Omega\left(k(\epsilon_1 - \epsilon_2)^2\right) - O(\log(h+k)).$$

### 3.2. From DP Streaming Algorithm to Communication Protocol

Having established a communication lower bound for the AvoidHeavyHitters game, we make a connection between it and DP streaming algorithm for CountDistinct.

Recall from Section 2.3 that $r(i, j)$'s are the answers produced by the algorithm. The following main lemma shows how to convert accurate answers for CountDistinct into sets $Y_i$ satisfying the constraints for AvoidHeavyHitters:

**Lemma 3.2.** *Suppose that the algorithm $A$ satisfies accuracy guarantee as in Theorem 2.1. Then, there exists a procedure Rounding that takes in $r(i, 1), \ldots, r(i, w), S \cup C_i$, and produces $Y_i$ such that the following holds with probability $1 - \frac{1}{10P^2}$: $|Y_i| \geq \left(\frac{1}{2} + \epsilon_1\right) \cdot |S_i \cup C|$ and there is no $u \in C$ that appears in more than $\left(\frac{1}{2} + \epsilon_2\right) P$ submitted subsets for $\epsilon_1 = \Theta\left(\frac{\sqrt{w}}{h\sqrt{\log(Ph)}}\right), \epsilon_2 = O\left(\frac{1}{\sqrt{P}}\right).$*

Before we describe this "rounding" procedure, let us first note that it can be easily turned into a communication protocol by letting player simulates the algorithm on one phase of the stream, as formalized below.

**Corollary 3.3.** *Let the parameters be as in Lemma 3.2. Suppose that the algorithm $A$ satisfies accuracy guarantee as in Theorem 2.1, and let $M$ denote its space complexity. Then, there is a protocol with communication complexity $M$ that wins AvoidHeavyHitters with probability $1 - \frac{1}{10P}$.*

*Proof.* For the $i$-th player, the protocol works as follows:

- The player receives the memory configuration of the algorithm from the $(i-1)$-th player (if $i > 1$), as well as the set $S_i \cup C$ from the referee.

- The player then *constructs* the $i$-th phase of the stream as described in Section 2.3: Importantly, the player can construct the inputs only with the knowledge of $S_i \cup C$, without telling $S_i$ apart from $C$.

- The player then feeds the updates to $A$ and collects responses $r(i, 1), \ldots, r(i, w)$ from it. After that, the player uses Rounding to compute $Y_i \subseteq S_i \cup C$.

- If $|Y_i| < \left(\frac{1}{2} + \epsilon_1\right)|S_i \cup C|$, the player aborts the protocol (and loses). Otherwise, the player sends $Y_i$ to the referee, and the memory configuration to the next player (if $i < P$).

By Lemma 3.2 and the union bound over all phases, this protocol wins the game with probability $1 - \frac{1}{10P}$. We remark that in the reduction, we only use the space bound of $A$ to upper-bound the length of the messages the players send and receive. In particular, to make up the queries, collect and process the responses, the players are allowed to use arbitrarily large working memory and time. $\square$

#### 3.2.1. CORRELATION ANALYSIS AND ROUNDING ALGORITHM

The remainder of this subsection is devoted to the description and analysis of Rounding.

**Analysis overview.** We will be studying the so-called "score statistics" of each user. Recall from Section 2.3 that for each user $u$, each phase $i$ and repetition $j$ we defined $p(i, j)$, $b(u, i, j)$ and $a(i, j)$. We then define the "score" of user $u$ in one phase as

$$\xi_u^{(i)} = \sum_{j \in [w]} a(i, j) \cdot (b(u, i, j) - p(i, j)). \quad (1)$$

Also, define the total score across all phases as

$$\xi_u = \sum_{i \in [P], j \in [w]} a(i, j) \cdot (b(u, i, j) - p(i, j)). \quad (2)$$

As we will demonstrate momentarily, the score statistics allow us to connect the privacy and accuracy properties of the streaming algorithm and prove Lemma 3.2. Specifically:

- In the first part of the section, we use the Fingerprinting lemma to prove that $\sum_u \xi_u$ is "large". Statistically we see the the algorithm's output is trying to estimate $p(i, j)$ (indeed, if the algorithm was unbiased, the expectation of $a(i, j)$ would be exactly $p(i, j)$). However,

since the algorithm cannot access $p(i, j)$ directly, and it produces the estimate via $b(u, i, j)$, the biases of the outputs are usually "aligned" with the biases of $b(u, i, j)$. By the Fingerprinting lemma, we can make a quantitative statement and give sharp lower bounds on the sum of scores (equivalently, the correlation between $b$ and $a$).

- In the second part, we conduct the following thought experiment: imagine we sample $b'(u, i, j)$ independently from $\mathrm{Ber}(p(i, j))$. Then, we define the shadow correlation $\xi'_u = \sum a(i, j)(b'(u, i, j) - p(i, j))$. The shadow score can be shown to be small and well-concentrated because $b'(u, i, j)$ has absolutely no correlation with $a(i, j)$. However, assuming the algorithm is differentially private, its output distribution shouldn't change too much if we change the input from $b$ to $b'$. Equivalently, the distribution $\xi_u$ and $\xi'_u$ is $(\epsilon, \delta)$-indistinguishable (in the DP sense).

Once we have established the necessary properties about $\xi_u$, we will use the scores to sample the subsets $Y_i$, via a procedure that we call "rounding". The statistical properties of $\xi_u$ translate to the properties of the $Y_i$'s, as claimed in Lemma 3.2. In the following, we give more quantitative details, with a full treatment deferred to the appendix.

**Fingerprinting Lemma.** The main ingredient required for Rounding is the well-known fingerprinting lemma:

**Lemma 3.4** (See, e.g., Bun et al. (2018); Dwork et al. (2015)). *Let $f : \{0, 1\}^n \to [0, 1]$ be a function. Suppose for every $x \in \{0, 1\}^n$, it holds that $|f(x) - \frac{1}{n} \sum_i x_i| \le \frac{2}{5}$. Then, it holds that*

$$\mathbb{E}_{p \in [0,1]} \mathbb{E}_{x \sim \mathrm{Ber}(p)^n} \left[ f(x) \sum_{1 \le i \le n} (x_i - p) \right] \ge \frac{1}{10}. \quad (3)$$

This simple yet powerful lemma is typically interpreted as follows. Imagine $f$ is a mean-estimator: it receives $n$ inputs bits and approximates their mean. The assumption of Lemma 3.4 says that $f$ should be a somewhat accurate estimator. The conclusion of the lemma, on the other hand, implies that *any* such estimator must have a noticeably high correlation with its inputs.

**Measuring correlation.** Let $\beta = \frac{1}{T^2}$ be a small parameter for analysis. Looking into the construction of Section 2.3, let $u \in C$ be a heavy user. Consider the following thought experiment: we sample independently $b'(u, i, j)$ in the same way as we sample $b(u, i, j)$. Then, the quantity

$$\xi' = \sum_{i \in [P], j \in [w]} a(i, j) \cdot (b'(u, i, j) - p(i, j)) \quad (4)$$

can be understood as the summation of $P \cdot w$ bounded (in $[-1, 1]$), mean-zero and independent random variables

(over the randomness $p(i, j), a(i, j), b'(u, i, j)$). As such, it is easily seen to be $(\sqrt{Pw})$-subgaussian[5] and we have that $\Pr[\xi' \le \sqrt{Pw \log(2/\beta)}] \ge 1 - \beta$.

If the algorithm is $(\epsilon, \delta)$-DP, it should not depend strongly on $b(u, i, j)$'s for any single user $u$. This is to say, the output distribution of the algorithm should not change too much when we change the input[6] from $b(u, i, j)$ to $b'(u, i, j)$. Consequently, define the random variable $\xi = \sum_{i,j} a(i, j) \cdot (b(u, i, j) - p(i, j))$, we have

$$\begin{aligned}
&\Pr[\xi > \sqrt{Pw \log(2/\beta)}] \\
&\le e^\varepsilon \Pr[\xi' > \sqrt{Pw \log(2/\beta)}] + \delta \\
&< 2\beta + \delta.
\end{aligned}$$

Similarly, for any $i \in [P]$ and a light user $u \in S_i$, the quantity $\sum_{j \in [w]} a(i, j)(b(u, i, j) - p(i, j))$ is bounded in absolute value by $\sqrt{w \log(2/\beta)}$ with probability $1 - 2\beta - \delta$.

**The Rounding Procedure.** We are now ready to introduce the rounding trick: after the conclusion of one stage $i \in [P]$, we can measure the quantity

$$\sum_{u \in S_i \cup C} \sum_{j \in [w]} a(i, j) \cdot (b(u, i, j) - p(i, j)). \quad (5)$$

For each $u \in S_i \cup C$, the inner sum is bounded in absolute value by $R = \sqrt{w \log(2/\beta)}$ with probability $1 - O(\beta + \delta)$. We condition on the event that all the inner sums are bounded by $R$, and we sample a random set $Y_i$ as follows: for each $u \in S_i \cup C$, include $u$ in $Y_i$ with probability $\frac{R + \sum_{j \in [w]} a(i,j) \cdot (b(u,i,j) - p(i,j))}{2R} \in [0, 1]$.

As it will turn out, from the sampling procedure we usually produces sets $Y_i$ of size $|S_i \cup C|(\frac{1}{2} + \widetilde{\Omega}(\frac{\sqrt{w}}{h}))$. To quickly grasp a quantitative intuition, note that if the algorithm is accurate, we can use Lemma 3.4 to deduce that (5) is lower bounded in expectation by $\Omega(w)$ (as Equation (5) corresponds to $w$ independent queries). That means the size of the sampled subset deviates from $\frac{1}{2}$ by $\frac{w}{R \cdot (3h+k)} \approx \frac{\sqrt{w}}{h}$. On the other hand, by the DP property of the algorithm, for each heavy user $u \in C$ the total correlation score it received (as in (4)) is at upper bounded by $\sqrt{Pw \log(2/\beta)}$. As such, in expectation it appears in at most $\frac{1}{2}P + \frac{\sqrt{Pw \log(2/\beta)}}{R} = \frac{1}{2}P + O(\sqrt{P})$ submissions. See Appendix A.2 on how to convert the in-expectation calculation into a high-probability statements.

---

[5]See Appendix A.1 for definition and properties about subgaussian r.v.s.

[6]Technically, $b(u, i, j)$ does not directly constitute the input stream. However, from the construction we see that the inputs are constructed directly from $b(u, i, j)$'s. Therefore, we think of $b$ as the "true hidden input" to the algorithm.

### 3.3. Proof of Theorem 2.1

Theorem 2.1 follows from Lemma 3.1 and Corollary 3.3. Recall that $\epsilon_1 = \Theta\left(\frac{\sqrt{w}}{h\sqrt{\log(Ph)}}\right), \epsilon_2 = O\left(\frac{1}{\sqrt{P}}\right)$ from Corollary 3.3. For the parameters in Theorem 2.1, we have

$$\frac{\epsilon_1}{\epsilon_2} \geq \Omega\left(\frac{\sqrt{w} \cdot \sqrt{T}}{h \cdot \sqrt{\log T} \cdot \sqrt{w(3h+k)}}\right) \geq 2.$$

Therefore, by Lemma 3.1, we obtain the lower bound

$$M \geq \widetilde{\Omega}(k(\epsilon_1 - \epsilon_2)^2) = \widetilde{\Omega}\left(\frac{kw}{h^2}\right).$$

## 4. Conclusions

In this work, we established the first unconditional space lower bounds for user-level differentially private streaming algorithms. This addresses a recent open problem by demonstrating that the polynomial space usage of existing private algorithms is essentially optimal (in certain regimes of CountDistinct) and establishes an exponential separation compared to non-private counterparts. Our proof hinges on a new communication game that shows that the memory-intensive task of tracking "over-active" users is fundamentally necessary for privacy in many instances. Furthermore, we showed this framework generalizes to problems like MaxSelect and Quantile estimation, indicating that high memory costs are inherent to a broad class of private continuous monitoring tasks. We believe our proof technique can be applied to many other problems that have so far escaped low space private algorithms.

## Impact statement

This paper presents work whose goal is to advance the field of machine learning. There are many potential societal consequences of our work, none of which we feel must be specifically highlighted here.

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

# Appendix

## A. Missing Proofs from Section 3

### A.1. Technical preliminaries

**Definition A.1** (Subgaussian r.v.)**.** A real-valued random variable $X$ is called $b$-subgaussian, if $\Pr[|X| > t] \leq 2\exp(-t^2/b^2)$ for every $t > 0$.

**Lemma A.2** (Sum of Subgaussian r.v.'s)**.** *Suppose* $X_1, \ldots, X_N$ *are* $N$ *independent random variables. If* $X_i$ *is* $b_i$-*subgaussian for* $i \in [N]$, *then* $\sum X_i$ *is* $\sqrt{\sum b_i{}^2}$-*subgaussian.*

**Lemma A.3** (Azuma's inequality)**.** *Suppose* $X_0, X_1, \ldots, X_N$ *is a submartingale[7], such that* $|X_i - X_{i-1}| \leq B$ *almost surely. Then*

$$\Pr[X_N - X_0 \leq -t] \leq \exp\left(-\frac{t^2}{2NB^2}\right).$$

A convenient variant of Lemma A.3 that we use is recorded below.

**Lemma A.4.** *Suppose* $X_0, X_1, \ldots, X_N$ *is a submartingale such that* $|X_i - X_{i-1}| \leq B$ *almost surely, and* $\mathbb{E}[X_{i+1} \mid X_i] \geq X_i + c$. *Suppose* $(X'_0, X'_1, \ldots, X'_N)$ *is a list of random variables such that* $d_{\mathrm{TV}}((X_i)_{0 \leq i \leq N}; (X'_i)_{0 \leq i \leq N}) \leq \beta$. *Then*

$$\Pr[X'_N - X'_0 \geq cN - t] \geq 1 - \exp\left(-\frac{t^2}{2NB^2}\right) - \beta.$$

*Proof.* Up to the TV distance of $\beta$ we can switch from $X'_i$'s to $X_i$'s. In order to analyze $X_0, \ldots, X_N$ we define $Y_i = X_i - i \cdot c$ and make use of Lemma A.3. $\square$

### A.2. The rounding analysis

Continuing our discussion in Section 3.2, we now lower-bound the size of the sampled subsets. Recall that each player $i \in [P]$ will construct $w$ queries to the streaming algorithm.

Consider the $j$-th query of player $i$. We define the history of the algorithm up to before the $j$-th query as the interaction between the algorithm and the first $(i - 1)$ players, as well as the first $(j - 1)$ queries by the current player $i$.

For each $1 \leq j \leq w$, we say the algorithm is *useful* for the $j$-th query of player $i$, if conditioned on the history up to before the query $j$, for every $p(i, j) \in [0, 1]$ that could have been sampled, it holds that

$$\left| \mathop{\mathbb{E}}_{b(u,i,j)\sim\mathrm{Ber}(p(i,j))}[a(i,j) \mid \text{history}] - p(i,j) \right| \leq \frac{2}{5}.$$

---

[7] Recall that $(X_0, \ldots, X_N)$ is a submartingale if $\mathbb{E}[X_{i+1} \mid X_i] \geq X_i$ for every $0 \leq i \leq N - 1$.

Colloquially, the algorithm is useful for the $j$-th query if, before the algorithm is presented the $j$-th query, its internal state such that it does not fails to answer queries accurately. Two remarks are in order about the definition: first, since the players define the queries and simulate the algorithm in place, they can check for usefulness given the current memory configuration of the algorithm. Second, if the algorithm always answers the queries to within the $(\eta = 1 + \frac{1}{10}, \tau = h)$-approximation guarantee, the algorithm would have been useful for every query: to see this, observe that in our construction we can convert the $(\eta, \tau)$-approximation guarantee to a $\frac{1}{10}(3h + k) + h$ additive error, which is below $\frac{2}{5}$ fraction of users from $S_i \cup C$.

We take "being useful for the $j$-th query" as an event depending on the history of the algorithm. By the accuracy guarantee of the algorithm, we know that the algorithm remains useful for every query with probability $1 - \frac{1}{T^2}$. Imagine a *hypothetical world*, where the players will convert the algorithm to an absolutely correct mean estimator whenever the algorithm is found to be not useful. In the hypothetical world, the algorithm will indeed be useful throughout. Moreover, the total variation distance between the real world and the hypothetical world is at most $\frac{1}{T^2}$.

Now, re-write (5) as

$$\sum_{j \in [w]} \left( \sum_{u \in S_i \cup C} a(i,j) \cdot (b(u,i,j) - p(i,j)) \right). \quad (6)$$

This is a sum of $w$ terms where each query contributes one term. In the *hypothetical world*, by Lemma 3.4, condition on the first $(j - 1)$ terms, the next term is bounded by $[-\sqrt{5h\log(1/10\beta)}, \sqrt{5h\log(1/10\beta)}]$ and has mean $\frac{1}{10}$. We monitor the partial sum of (6) as $j$ increases. This is clearly a submartingale, and we can make use of Azuma's inequality (we use the variant from Lemma A.4) to deduce for the *real world* that

$$\Pr\left[ \sum_{j \in [w]} \left( \sum_{u \in S_i \cup C} a(i,j) \cdot (b(u,i,j) - p(i,j)) \right) > \frac{w}{20} \right]$$
$$\geq 1 - \exp\left( -\Omega\left( \frac{w^2}{w \cdot h \cdot \log(1/\beta)} \right) \right) - \frac{1}{T^2}$$
$$\geq 1 - \exp(-\Omega(w/h)) - \frac{1}{T^2}$$
$$\geq 1 - O(\beta).$$

Therefore, we see that on average each inner sum of Equation (5) is $\frac{w}{100h}$. By the choice of $R$, we see that the result of the sampling procedure produces a set $Y_i \subseteq S_i \cup C$ of expected size $|S_i \cup C| \left( \frac{1}{2} + \Theta\left( \frac{\sqrt{w}}{h\sqrt{\log(1/\beta)}} \right) \right)$. Using

concentration inequality, we obtain that the size of $Y_i$ is lower bounded by $\left(\frac{1}{2} + \frac{1}{2}\Theta\left(\frac{\sqrt{w}}{h\sqrt{\log(1/\beta)}}\right)\right)$ with probability $1 - \exp\left(-\Omega(\frac{w \cdot h}{h^2 \log(1/\beta)})\right) \geq 1 - \beta$.

On the other hand, by the privacy property of the algorithm, we have, for every $u \in C$, that the bound $\sum_i \sum_j a(i,j) \cdot (b(u,i,j) - p(i,j)) \leq \sqrt{Pw \log(1/\beta)}$ holds with probability $1 - 2\delta + \beta$. By our choice of $R = \sqrt{w \log(2/\beta)}$, it follows that with probability $1 - O(\beta)$ the user $u$ appears in at most $\frac{1}{2}P + O(\sqrt{P})$ many $Y_i$'s, by our choice of $R$.

By our choice of $\beta = \frac{1}{T^2}$, we can afford to union-bound over all sets $Y_i$, and deduce that the sets $Y_i$ satisfy the requirement of Lemma 3.2 with probability at least $1 - \frac{1}{T} \geq 1 - \frac{1}{10Ph}$.

### A.3. Proof of Lemma 3.1

To prove Lemma 3.1, we make use of the following claim (proved later).

**Lemma A.5.** *Let* $(A, m) \in \binom{[N]}{k} \times \{0,1\}^M$ *be a joint random variable such that the marginal distribution of $A$ is uniform over $k$-subsets of $[N]$. Suppose $f : \{0,1\}^M \to \Delta\left(\binom{[N]}{k'}\right)$ is a randomized[8] mapping that takes input $m \in \{0,1\}^M$ and outputs a $k'$-subset of $[N]$. Denote $Z = \mathbb{E}_{A,m,f(m)}[|f(m) \cap A|]$. Then, it holds that*

$$M \geq \log\binom{N}{k} - \log k - \log\binom{k'}{Z} - \log\binom{N - k'}{k - Z}$$

Assuming the claim, we prove Lemma 3.1.

*Proof of Lemma 3.1.* We can assume that every player always sends in a subset of size exactly $\left(\frac{1}{2} + \epsilon_1\right)|S_i \cup C|$. By our assumption, every $u \in C$ appears in no more than $\left(\frac{1}{2} + \epsilon_2\right)P$ submitted sets with probability $1 - \frac{1}{10P}$. It follows that in expectation every $u \in C$ appears in less than $\left(\frac{1}{2} + \epsilon_2\right)P + 1$ submissions. As there are $P$ players in total, it follows that there is at least one player $i$ such that the subset $Y_i$ from that player satisfies that $\mathbb{E}[|Y_i \cap C|] \leq \left(\frac{1}{2} + \epsilon_2\right)k$. Fix one such player $i^*$.

We know that Player $i^*$ produces their subset $Y_{i^*}$ from the message $m_{i^*-1}$ and the ground set $S_{i^*} \cup C$. Moreover, the set $Y_{i^*}$ has the property that $|Y_{i^*}|$ is fixed to $\left(\frac{1}{2}+\epsilon_1\right)(3h+k)$ and $\mathbb{E}[|Y_{i^*} \cap C|] \leq \left(\frac{1}{2} + \epsilon_2\right)k$. We then apply Lemma A.5 to $i^*$ to deduce that

$$M \geq \log\binom{3h+k}{k} - \log\binom{(3h+k)(\frac{1}{2} + \epsilon_1)}{\left(\frac{1}{2} + \epsilon_2\right)k}$$
$$- \log\binom{(3h+k)(\frac{1}{2} - \epsilon_1)}{\left(\frac{1}{2} - \epsilon_2\right)k} - O\left(\log(h+k)\right),$$

---

[8]We use $\Delta(U)$ to denote the probability simplex over a set $U$. We now establish the bonus part of Lemma 3.1: i.e., how we arrive at the final, cleaner expression via Stirling's approximation. Write $P = 3h + k$, $Q = k$, and $P' = \left(\frac{1}{2} + \epsilon_1\right)P$,

$Q' = \left(\frac{1}{2} + \epsilon_2\right)Q$ for brevity. The first three terms on the right hand side may be written as

$$\log\binom{P}{Q} - \log\binom{P'}{Q'} - \log\binom{P - P'}{Q - Q'}.$$

Given a pair $(N, M)$, write $\alpha = \frac{M}{N}$ and one has $\ln\binom{N}{M} = N \cdot H(\alpha) \pm O(\ln N)$. This allows us to write:

$$\log\binom{P}{Q} - \log\binom{P'}{Q'} - \log\binom{P - P'}{Q - Q'}$$
$$= P \cdot H(\frac{Q}{P}) - P'H(\frac{Q'}{P'}) - (P - P')H(\frac{Q - Q'}{P - P'})$$
$$\pm O(\ln P).$$

After a routine calculation, the first three terms may simplify to $P \cdot I(X \mid Y)$ where $X, Y$ are joint binary random variables such that $\Pr[X = 1] = \frac{Q}{P}, \Pr[Y = 1] = \frac{1}{2} + \epsilon_1$ and $\Pr[Y = 1 \mid X = 1] = \frac{1}{2} + \epsilon_2$. The term $I(X \mid Y)$ can be shown to be at least $\frac{Q}{P}D(\text{Ber}(b)\|\text{Ber}(a))$. After simplification, we get the desired conclusion of $\Omega(k(\epsilon_1 - \epsilon_2)^2)$. $\square$

We give the proof of Lemma A.5 below.

*Proof of Lemma A.5.* We use an encoding argument. First, denote $B = f(m)$ and let us consider the Markov chain $A \to m \to B$. We now encode the set $A$ given $B$, using some additional side information $C$ that we introduce below. First, let $I := |A \cap B|$. Then, describe the set $A \cap B$ and $A \setminus B$ *with reference* to $I$ and $B$ (that means $A \setminus B$ is represented as a subset of $[N] \setminus B$, and $A \cap B$ a subset of $B$). Finally, define $C$ as the tuple $(I, A \cap B, A \setminus B)$.

We shall now consider the joint random variable $(A, m, B, C)$. We see that $H(B, C) \geq H(A)$ because $A$ can be recovered exactly given only $B$ and $C$. We also have $H(B) \leq H(m) \leq M$ by the data processing inequality. Lastly we know that $H(B, C) = H(B) + H(C \mid B)$. Taken together, we obtain

$$M + H(C \mid B) \geq H(B) + H(C \mid B) \geq H(A) = \log\binom{N}{k}.$$

So it remains to give an upper bound for $H(C \mid B)$. Denote $I = |A \cap B|$. We have

$$H(C \mid B) \leq H(I \mid B) + H(A \mid B, I)$$
$$\leq \log(k) + \mathbb{E}_I\left[\log\binom{|B|}{I} + \log\binom{N - |B|}{k - I}\right]$$
$$\leq \log(k) + \log\binom{|B|}{Z} + \log\binom{N - |B|}{k - Z},$$

where the last inequality is due to the log-concavity of binomial coefficients. The conclusion now follows by plugging the upper bound of $H(C)$ into the inequality above. $\square$

## A.4. Proof of Theorem 2.3

Here we describe how to extend the proofs to other problems. In order to prove Theorem 2.3, we will replace the fingerprinting lemma (Lemma 3.4) with a recent "strong" variant by (Peter et al., 2024). Formally, their lemma reads:

**Lemma A.6** (Strong fingerprinting lemma of (Peter et al., 2024)). *Let $f : \{0,1\}^n \to [0,1]$ be a function, such that $f(0^n) \le 0.1$ and $f(1^n) \ge 0.9$. Then, it holds that*

$$\mathop{\mathbb{E}}_{\substack{t \sim [-\log(5n), \log(5n)] \\ p := \frac{e^t}{e^t+1}}} \mathop{\mathbb{E}}_{x \sim \mathrm{Ber}(p)^n} \left[ f(x) \sum_{1 \le i \le n} (x_i - p) \right]$$

$$\ge \Omega\left(\frac{1}{\log(n)}\right). \tag{7}$$

Let us quickly compare Lemma A.6 with Lemma 3.4. The most significant advantage of Lemma A.6 (why it is called "strong") is that we have a much more relaxed requirement for the function $f$: namely, when viewed as an empirical mean estimator, we only need $f$ to be accurate on two extreme inputs $0^n$ and $1^n$. The "price" we pay is that we need to change how the prior $p$ is defined, and the conclusion we obtain is weaker than Lemma 3.4 by a logarithmic factor. However, since we are focusing the polynomial factors in proving lower bounds, we can afford to lose a "log" here.

Equipped with Lemma A.6, it remains to design a "mean estimator" from a Quantile or MaxSelect streaming algorithm. Recall this was very easy for the case of CountDistinct: we only insert all users that "contribute" to a query into the stream. By asking for the current distinct count, we can infer how many users have contributed to the query. For the new problems, we will use the following gadget instead:

- Quantile: Suppose there is a set $Q$ of active users out of all $[N]$ users, where each active user holds a private bit, and we are interested in distinguishing whether they all hold 0 or they all hold 1 (if there is no consensus, our estimator is allowed to output anything). We create a universe $U$ with two items, namely $U = \{0, 1\}$. Then we assign each active user an item corresponding to their private bit. For the users from $[N] \setminus Q$, we simply set their item to 1.

  Now, consider querying for the rank-$\frac{Q}{2}$ item among all users. If the algorithm has additive error less than $\frac{Q}{2}$, we see that the algorithm has to output the same bit if all active users hold that bit. This is exactly what is required in Lemma A.6.

- MaxSelect: Again, suppose there are $Q$ active users each holding a private bit. Create two features (i.e.,

$d = 2$) for the MaxSelect problem, and we set the first feature as one for half of all users (the selected half is randomly chosen and is considered a piece of public information). Then for every active user with a "one" input, we set the second feature of that user as one. We set the feature vectors of inactive users to be all-zero.

Now we query for the more popular feature among the two. Assuming the algorithm has additive error less than $\frac{Q}{2}$, we see that the algorithm has to output Feature 1 if none of the users is in favor of the second feature. On the other hand, the algorithm needs to output Feature 2 if all of them are in favor of the second feature. Thus, this reduction also gives an "estimator" matching the requirement of Lemma A.6.

**Construction overview.** Given these gadgets, one can easily modify the construction of Section 2.3 to perform a reduction from "answering one-way marginal queries" to "answering Quantile or MaxSelect queries". In slightly more detail, given the parameters $w, k, h, T$, we generate the sets $C, S_1, \ldots, S_P$ in the same way as in Section 2.3. Then we design the input stream: in Step (1) we will sample $p$ according to the rule given by Lemma A.6 (namely, set $n = 3h + k$, choose $t \sim [-\log(5n), \log(5n)]$ and set $p = \frac{e^t}{e^t+1}$). In Step (2) we create updates according to the rules specified above. In Step (3) we make a query, whose answer can be converted into a bit. Lastly, in Step (4) we undo the updates we did in Step (2).

**Analysis overview.** Given the construction, we apply a similar analysis as those appeared in Section 3.2 and Section 3.1. Namely, we can define the correlation terms, prove that they are bounded with high probability by the privacy property of the algorithm. At the same time, the correlation terms are lower bounded by the new Fingerprinting lemma of Lemma A.6. We then use the correlation terms to sample subsets, which induces a communication protocol for AvoidHeavyHitters as in Section 3.1. Finally, the desired space lower bound readily follows from the established lower bounds for AvoidHeavyHitters. The quantitative aspect of the new analysis remains essentially the same as before, except for one minor detail: namely the correlation lower bound we obtained from Lemma A.6 is weaker than the one from Lemma 3.4 by a $\log(n)$ factor. As a result, the parameter $\epsilon_1$ for AvoidHeavyHitters gets smaller by a factor of $\log(n)$, which ultimately makes our lower bound weaker by a factor of $\log^2(n)$. Still, we emphasize that this does not affect our main result of Theorem 2.3, as the logarithmic factors can be "absorbed" into the $\widetilde{O}$ notation.

