# OpenReview forum: "Keeping a Secret Requires a Good Memory: Unconditional Streaming Lower-Bounds for Differentially Private Algorithms"
_ICML.cc/2026/Conference — ICML 2026 spotlight_

### Official Review · Reviewer_9MEP · 2026-02-23

**Soundness:** 3
**Presentation:** 2
**Significance:** 4
**Originality:** 4
**Overall Recommendation:** 5
**Confidence:** 4

**Summary:**

This paper derives the memory lower bound for user-level DP (eps is constant and delta is small enough) streaming counting algorithms (e.g. Distinct Count, MaxSelect and Quantile). This is the first such result without any hardness assumption and indeed shows the exponential separation between private and non-private algorithms (those sketching algorithms like HyperLogLog).
The main technical result is pretty clean: it shows that to achieve $o(T^{1/3})$ error, $\widetilde{\Omega}(T^{1/3-o(1)})$ space is necessary.

The proof technique is very nice: first, the authors constructed a hard instance for the occurrence-bounded version of the DistinctCount problem as follows: there's at most k heavy-updating users and n-k light-updating users (no more than w updates). The light users are grouped into P disjoint groups. Then, for each group, the constructed stream will comprise the updates only including the particular light group and the k heavy users, and the algorithm is required to answer the one-way marginal query. It will be repeated w times.
They then proceed to consider a very interesting communication game -- each player i can observe the unordered joint set of $P_i \cup C$ where $C$ is the heavy users. They can receive at most M bits from the last player (the memory trace). There are two main winning conditions: 1) They are required to guess a subset that's larger than $(1/2+\epsilon_1) |P_i \cup C| $; 2) no heavy users are included more than $(1/2+\epsilon_2)P$ times where $\epsilon_2<\epsilon_1$.

Given this hard instance as the centerpiece, there are two major steps towards the proof:
1) Show a memory lower bound for this game -- the transmitted bits have to carry enough entropy for the players to hit the target (as $\epsilon_2<\epsilon_1$) -- a very standard entropy counting argument.
2) Show that a user-level DP stream counting algorithm with sufficient correctness yields a winning strategy -- each player can sample the indices based on their "correlation score" to the DP counting outputs. This provides two key features: the correctness guarantee of the streaming counting algorithm with the fingerprinting lemma shows the output set will be big enough, while the DP guarantee says that the correlation score of any heavy user is bounded because we can always switch the heavy users to dummies and the output distribution is still eps-delta close.

Overall, I find the paper provides a very insightful new perspective in proving memory lower bounds for DP algorithms and should be considered a very strong submission.

**Compliance With Llm Reviewing Policy:**

Affirmed.

**Final Justification:**

A strong paper and should be accepted.

**Key Questions For Authors:**

Only minor issues:

1. Please highlight where DP is used in the proof process -- I find it not clear enough in the first place. It is actually used to break the dependency between a(i,j) and b(u,i,j) so that the correlation score of those heavy users can be bounded. I wish this can be explained more clearly for other readers.

2. The title should include "Space Lower-Bounds for Private Algorithms" -- Private Algorithms could mean a lot of things and there are so many other privacy definitions. A better title could be "Unconditional Space Lower-Bounds for Differentially Private Streaming Countings."

3. The Strealing approximation process is missing.

4. In "proof of lemma A.5" in appendix A.3, why do you need the I=|A \cap B| in the calculation?

5. Line 348 -- Intuitively.... eps_2 > eps_1 ---> wrong direction?

**Limitations:**

yes

**Strengths And Weaknesses:**

Good:

1. The result itself is strong enough. It's the first IT lower bound in this space, showing the actual exponential separation between private and non-private algorithms in terms of space, and also proves some existing constructions are already near-optimal.

2. The proof technique is really elegant and provides new perspective in proving more memory lower bounds for DP algorithms.

Bad:

I am very familiar with DP streaming algorithms, but it still takes me quite a long time figure out the whole flow (maybe I am not that familiar with lower bound proving techniques). The presentation could be done better (e.g. use roadmap figures to clearly demonstrate the proof technique, providing more insights, etc.)

---

> ### Author Rebuttal · Authors · 2026-03-30
>
> Thank you for your time and effort in reviewing, and for the constructive feedback!
>
> 1. "highlight where DP is used in the proof process": Thank you for the suggestion. Indeed, currently the DP assumption is used deep in Section 3.2.1, where we formally described and analyzed the "rounding" procedure, making it harder to grasp the intuition. In the next revision, we will add a gentle introduction at the beginning of Section 3.2.1, defining the correlation scores and describing how they are high on average (by the fingerprinting lemma) and low on heavy-hitters (by the DP property of the algorithm). Only after these, we will present the quantitative details.
> 2. Regarding the title: thank you for the suggestion. We will edit the title to reflect that we work with the space complexity of differentially private streaming problems. The problem we considered included both private counting and some induced optimization/inference tasks such as quantile and the so-called MaxSelect problem. Conceptually, our paper aims to describe a source of "space bottleneck" that is broadly applicable to DP streaming problems.
> 3. The Stirling approximation: We will add the formal approximation in the formal proof.
> 4. Re proof of Lemma A.5: By recording $I = |A\cap B|$, we can condition on the size of the intersection and conveniently work with subsets of a fixed size. The price of conditioning on $I$ is only $\log(k)$, which is minor in our setting. Without this conditioning, it seems more subtle to calculate the conditional entropy of $H(A|B)$.
> 5. Re Line 348: Yes, it should be $\varepsilon_1 > \varepsilon_2$. Thank you for pointing this out.

---

> > ### Author Rebuttal · Reviewer_9MEP · 2026-04-02
> >
> > Key issues addressed.

---

### Official Review · Reviewer_97ME · 2026-03-04

**Soundness:** 4
**Presentation:** 4
**Significance:** 3
**Originality:** 4
**Overall Recommendation:** 5
**Confidence:** 4

**Summary:**

This paper provides a lower bound on the *space* needed for a collections of relatively natural differential privacy problems over streams.  The exact statements are somewhat technical, but the very high-level interpretation is that *polynomial* space is needed to get "reasonable" accuracy for these problems under $(\epsilon, \delta)$-DP.  There are already known DP algorithm for these problem which do indeed achieve reasonable accuracy using polynomial space (although in most regimes the known upper bound does not exactly match the new lower bound), and it was also known how to get *polylogarithmic* space with the same basic accuracy if we don't care about privacy.  So this paper proves an *exponential* gap between the space needed for private algorithms vs non-private algorithms for a set of quite natural problems.  While this is not the first space lower bound for private algorithms, it is actually the second -- there is (to the best of my knowledge) one previous such lower bound, which this paper cites.  But this lower bound is for a non-natural problem, and depends on cryptographic assumptions.  This new lower bound is for a much more natural set of problems and is completely unconditional.

**Compliance With Llm Reviewing Policy:**

Affirmed.

**Final Justification:**

The rebuttal essentially agreed with my review -- there are some minor weaknesses, but this is overall a strong paper that should be accepted.

**Key Questions For Authors:**

I like this paper and think it should be accepted, so I do not have any questions.

**Limitations:**

Yes

**Strengths And Weaknesses:**

Strengths:
- There has been a significant amount of work on DP streaming algorithms, and in the non-DP setting most streaming work focuses on the space.  But this is essentially the first space lower bound for DP streaming.  This feels like a pretty major contribution to me.
- The problems that it considers are very natural, and are well-studied in both the non-DP and DP literature.  This further motivates the lower bound -- it is not contrived just to show that there exists some problem with a strong space cost of privacy, but actually gives a a natural problem.
- I did not check the math line by line, but it seems correct to me, and the basic proof structure and intuition is explained quite well.
- While there are many known lower bounds for DP, since this one is about space it required some new techniques.  It's not just the usual "packing lower bound" approach (or something similar) with some tweaks -- it seems genuinely new (at least to me).  So this seems quite original and impactful.

Weaknesses:
- While the problems are natural, the lower bound strongly depends on user-level privacy.  This is a very strong notion of privacy, so it is maybe not so surprising that there are strong lower bounds.  Ideally there would be a similar lower bound on event-level privacy.
- This paper doesn't really tell the "complete story" for the problems it considers.  There are some very limited regimes where the new lower bounds match the known upper bounds, but these are quite limited.  Generally there is still a reasonably large gap.  Ideally there would also be improved upper bounds to close this gap.

---

> ### Author Rebuttal · Authors · 2026-03-30
>
> Thank you for your time and effort in reviewing! We have the following brief response on the weaknesses:
>
> * "The user-level privacy being too strong":
> > (1) motivation-wise, user-level differential privacy is arguably the gold standard for real-world use cases of differential privacy: the "event-level" privacy struggles to offer meaningful privacy protection for individuals, especially so over a long course of streaming interaction where users can make a lot of updates.
> >
> > (2) from a technical perspective, it is true that user-level DP is too strong from a worst-case consideration. Nevertheless, as has been repeatedly demonstrated in a series of works, in user-level DP one can often take advantage of the distributional and/or structure properties of the inputs to design surprising algorithms: in fact, our paper studies natural streaming setting where the "bounded occurence" phenomenon of (most) users can be exploited to design more accurate algorithm. Here, our paper delievers a strong message, implying that a large working space is required to facilitate user-level DP algorithm design.
> >
> > (3) let us remark that we became aware of the paper (https://arxiv.org/abs/2602.10360) after our submission, which showed that some of the lower bounds presented in our paper can indeed be bypassed when one considers the more relaxed notion of event-level privacy: it is implicit in our paper that $(\log^{10}(T), T^{\Omega(1)})$-approximation to the streaming distinct count is information-theoretically hard in the user-level privacy setting (in the main body, we presented our proof for $(1.01, T^{\Omega(1)})$-approximation. By using the strong fingerprinting lemma (Lemma A.6), the stronger lower bound claimed here can be proved), while such an approximation is shown to be achievable under the event-level privacy in the referenced paper.
>
> * "This paper doesn't really tell the "complete story" for the problems it considers" : Yes, as has been communicated in our paper, our lower bound is tight/meaningful only in certain parameter regimes and the story has not been fully complete yet. We hope our work can inspire more follow-up research.

---

> > ### Author Rebuttal · Reviewer_97ME · 2026-03-31
> >
> > The authors and I seem to basically agree on the weaknesses of this paper, and also agree that they are not major -- this paper should be accepted.

---

### Official Review · Reviewer_735n · 2026-03-13

**Soundness:** 4
**Presentation:** 4
**Significance:** 3
**Originality:** 4
**Overall Recommendation:** 5
**Confidence:** 3

**Summary:**

The paper considers memory lower bounds for differentially private algorithms under user-level privacy and continual release setting. Given a stream of updates of length $T$, where for each step a user contributes an update, the goal is to compute some quantity at each step privately. The running example used in the work is the Count Distinct problem where we want to count the number of users contributing non-zero values at time $t$. Assuming that at most $k$ users have more than $w$ updates ($(w,k)$-occurrency), the authors show that any $(\varepsilon,\delta)$-DP algorithm with additive error $h$ requires at least $\Omega(kw / h^2)$ space (under certain parameter regime). The authors also discuss extending the lower bound to the quantile and max-select problem.

**Compliance With Llm Reviewing Policy:**

Affirmed.

**Final Justification:**

Strong set of results and well-written paper, I strongly lean towards acceptance.

**Key Questions For Authors:**

No major questions, but a clarification question
In line 348, the left column, should the inequality be $\varepsilon_1 > \varepsilon_2$ ?

**Limitations:**

As mentioned and highlighted in the work, the lower bound only works under certain parameter regime. Nevertheless, the contributions of this work are significant in proving the first unconditional lower bounds for memory requirements, and the general communication game might find use in proving lower bounds for additional problems.

**Strengths And Weaknesses:**

The paper is very well written and clearly motivated.
For the problem setting considered, this work obtains the first unconditional lower bounds on memory requirements for private algorithms. The authors introduce a novel multi-player communication game, Avoid Heavy Hitters, and show a lower bound the communication complexity for this problem. They then show that a low memory DP algorithm  implies a low communication protocol for this problem, thereby giving the memory lower bound for a private algorithm. Specifically, for the Count Distinct problem, for $(w,k)$-occurrency stream with appropriate values of $w,k,\alpha$ and length $T$ stream, they are able to derive $\Omega(T^{1/3 - 4\alpha})$ lower bound on memory for a DP algorithm with $O(T^{1/3-\alpha})$ additive error, which almost matches the upper bound of previous work. The theorem does not work for general arbitrary streams. Dependence on $\varepsilon$ for small and large range is unclear.

---

> ### Author Rebuttal · Authors · 2026-03-30
>
> Thank you for your time and effort in reviewing!
>
> Thank you for pointing out the typo: indeed, in Line 348 the inequality should be $\varepsilon_1 > \varepsilon_2$: intuitively the players need to guess large subsets (controlled by the parameter $\varepsilon_1$), while the heavy hitters cannot appear a lot of times (controlled by $\varepsilon_2$). So the commuinication game is challenging only when $\varepsilon_1 > \varepsilon_2$.
>
> And thank you for pointing out the limitation. Yes: as clearly communicated in our paper, our lower bound is tight/meaningful only in certain parameter regimes and the story has not been fully complete yet. We hope our work can inspire more follow-up research.

---

> > ### Author Rebuttal · Reviewer_735n · 2026-04-02
> >
> > All relevant concerns are addressed.

---

### Official Review · Reviewer_EvT3 · 2026-03-14

**Soundness:** 4
**Presentation:** 3
**Significance:** 4
**Originality:** 3
**Overall Recommendation:** 5
**Confidence:** 3

**Summary:**

This paper investigates the trade-off between privacy, utility, and space complexity in differentially private algorithms for continual statistics release of a stream under user-level differential privacy. It establishes a lower bound for the fundamental problem of counting distinct elements in a stream under the turnstile model. Specifically, for a stream of length $T$, any differentially private algorithm achieving accuracy $T^{1/3 - O(\alpha)}$ (with $\epsilon = 0.5$ and $\delta = 1/T^2$) must incur a space usage of at least $T^{1/3 - O(\alpha)}$ bits. This result demonstrates that the space complexity of the algorithm by Cummings et al. (2025) is essentially optimal and reveals an exponential gap in memory requirements compared to non-private counterparts.

**Compliance With Llm Reviewing Policy:**

Affirmed.

**Final Justification:**

The authors have fully addressed my question during the rebuttal phase. I maintain my positive assessment of this paper.

**Key Questions For Authors:**

Would it be possible to tighten the analysis if the concentration inequality in Lemma A.3 is replaced by a version that takes into consideration the variance of the random variables?

**Limitations:**

Yes

**Strengths And Weaknesses:**

- The paper establishes the first unconditional space lower bound for the problem.

- The paper resolves an open problem posed by (Cummings et al., 2025) on whether their space usage is tight.

- The paper establishes an exponential space usage separation between the private and non-private versions of the problem.

- The paper is relatively well-written.

---

> ### Author Rebuttal · Authors · 2026-03-30
>
> Thank you for your time and effort in reviewing! We are glad to see you liked the paper.
>
> Regarding the question, the answer is no: In our setting we used Lemma A.3 to analyze the sum of the correlation terms $a(i, j) \cdot (b(u, i, j) - p(i, j) )$. We have designed our constructions so that these terms are bounded by $[-1, 1]$ and their variances can be $\Omega(1)$. So, a variance-aware concentration inequality cannot help (at least not asymptotically) in our setting.

---

> > ### Author Rebuttal · Reviewer_EvT3 · 2026-04-03
> >
> > Thanks for answering my question.

---

### Decision · Program_Chairs · 2026-04-30

**Decision:**

Accept (spotlight)

**Comment:**

All reviewers recommend acceptance, agreeing that this paper makes a solid contribution by providing the first unconditional space lower bound for user-level differential privacy.  They found that the communication-game based proof technique elegant and novel, particularly for establishing a strong exponential separation between private and non-private streaming algorithms. While there were minor concerns regarding the specific parameter regimes and the clarity of certain sections, the reviewers felt the rebuttal sufficiently addressed these points and maintained their support for the work.